# Tensor network simulation of multi-environmental open quantum dynamics via machine learning and entanglement renormalisation

Florian A.Y.N. Schröder [1], David H.P. Turban[1], Andrew J. Musser[2], Nicholas D.M. Hine[3] & Alex W. Chin[4]

The simulation of open quantum dynamics is a critical tool for understanding how the non-classical properties of matter might be functionalised in future devices. However, unlocking the enormous potential of molecular quantum processes is highly challenging due to the very strong and non-Markovian coupling of 'environmental' molecular vibrations to the electronic 'system' degrees of freedom. Here, we present an advanced but general computational strategy that allows tensor network methods to effectively compute the non-perturbative, real-time dynamics of exponentially large vibronic wave functions of real molecules. We demonstrate how ab initio modelling, machine learning and entanglement analysis can enable simulations which provide real-time insight and direct visualisation of dissipative photophysics, and illustrate this with an example based on the ultrafast process known as singlet fission.

---

[1] Cavendish Laboratory, University of Cambridge, J. J. Thomson Avenue, Cambridge CB3 0HE, UK. [2] Department of Physics and Astronomy, University of Sheffield, Hounsfield Road, Sheffield S3 7RH, UK. [3] Department of Physics, University of Warwick, Gibbet Hill Road, Coventry CV4 7AL, UK. [4] CNRS & Institut des NanoSciences de Paris, Sorbonne Université, 4 place Jussieu, boite courrier 840, 75252 PARIS Cedex 05, France. Correspondence and requests for materials should be addressed to A.W.C. (email: alex.chin@insp.jussieu.fr)

Ultrafast open quantum dynamics have recently attracted considerable attention in the context of molecular light-harvesting materials, where experimental evidence has begun to highlight the role of coherent and non-equilibrium vibronic dynamics in systems ranging from organic photovoltaics to the pigment–protein complexes of photosynthesis[1,2]. Exploiting this complex interplay of vibrational and electronic quantum dynamics in tailored materials has been suggested to offer an exciting route towards advanced, next-generation light-harvesting devices[3,4], and the process known as singlet fission (SF) has recently been shown to be a very promising area in which to explore this. In this article, we present a powerful and general numerical approach designed to capture the full complexity of common molecular open system models, using the example of singlet fission to show how many-body simulation techniques can be combined with ab initio methods to elucidate and visualise the real-time sequence of system–environment processes that drive efficient light-harvesting processes.

In organic semiconductors, such as pentacene, a photo-generated singlet exciton undergoes singlet fission into a pair of entangled triplet excitons on sub-100 fs time scales, efficiently generating two electronic excitations from a single incoming photon[5,6]. Harnessing this carrier multiplication could help to overcome the Shockley–Queisser limit in photovoltaics[7], and understanding the links between the dynamics and efficiency of SF has emerged as an active interdisciplinary field of research[8–11]. Recently, ultrafast optical spectroscopy experiments have revealed how non-equilibrium and non-perturbative open quantum dynamics contribute to the kinetics of SF, highlighting the role of the molecular vibrational environment in driving ultrafast SF through rapid energy relaxation, conical intersections and vibronic mixing effects[12–16]. Importantly, these results exemplify the emerging idea that the evolving, high-dimensional quantum states of molecular environments can become quantum mechanically entangled with the fate of electronic photoexcitations, and that these correlations may transiently change electronic properties, such as spatial coherence and coupling to other parts of the environment or external probes[1,3,4]. These non-Markovian 'memory' effects could open novel pathways for controlling or optimising light-harvesting processes, but they arise from many-body physics which can only be explored with numerical techniques that move substantially beyond standard 'heat bath' treatments of the environment[6,17].

However, obtaining accurate real-time information about the environmental quantum state in open quantum systems is a major theoretical challenge, as it requires (1) the inclusion of a large number of vibrational modes or bath degrees of freedom to capture the effective irreversiblity of dissipative dynamics and (2) the simulation of vibronic wave functions whose sizes diverge exponentially in the number of vibrational modes. A powerful approach to this from chemical physics is the Multi-layer Multi-Configurational Time-Dependent Hartree (ML-MCTDH) algorithm[11,18–20], but, here, we draw upon the deep insights into many-body wave functions and the theory of low-rank tensor approximations that underlie the highly efficient matrix-product state (MPS) and tensor network state (TNS) ansatz used for strongly correlated problems in condensed matter[21–26]. Encouragingly, MPS ansätze has already been shown to provide highly accurate results for toy models of open systems with hundreds of quantised and highly excited vibrations[27–33], and have also very recently been applied to the equations of motion of reduced density matrices[34,35]. Unfortunately, MPS techniques are mostly effective for 1D systems, placing strong constraints on the types of the environment that can be simulated (vide infra). Molecular light-harvesting systems are characterised by much more structured vibronic couplings, and in order to profit from

the availability of ab initio parametrizations of molecular vibronic Hamiltonians, a higher-dimensional MPS-like method is required. While extensions to higher dimensions have proved challenging in condensed matter settings, we will demonstrate here that many molecular open system models, especially those with defined symmetry properties, can be efficiently simulated with higher-dimensional methods.

To acheive this, we introduce here a computational strategy based on tree tensor network states (TTNS), and give a step-by-step demonstration of how we can manipulate ab initio data with machine-learning techniques to formulate the problem in a form suitable for TTNS simulation. To evolve the resulting TTNS representation of the full many-body wave function, we employ a recent variational principle that allows fast and efficient processing[26,31,36], and—in a critical step—we show how an analysis of the entanglement 'topology' of the tensor network allows for a 'rewiring' of the network that can compress the memory requirements by up to six orders of magnitude. We shall then show that the formalism of TTNS enables a completely general 'on the fly' method to discover the dominant many-body/multidimensional configurations of the environment that drive open dynamics, and how this allows us to visualise non-equilibrium processes on just a few low-dimensional environmental energy surfaces. With this new capability to identify and distinguish effective 'reaction coordinates' according to their evolving relevance, we pinpoint the sequence of ultrafast vibrational motions that drive singlet fission in an ab initio parametreised dimer molecule known to undergo efficient SF, and highlight the potential utility of our technique for unravelling complex non-Markovian dynamics in a wide range of open quantum systems across physics and chemistry.

## Results

**Electronic structure and vibronic model.** We demonstrate our integrated computational strategy on an ab initio DFT-parameterised model of 13,13′-bis(mesityl)-6,6′-dipentacenyl (DP-Mes, Fig. 1a), a pentacene-based dimer molecule in which quantitative SF has been observed on ultrafast (sub-ps) time scales, despite SF being symmetry forbidden at its ground-state geometry[37,38]. Ab initio TD-DFT calculations have been performed on DP-Mes to derive a microscopic Hamiltonian model spanned by five electronic diabatic states, henceforth denoted as the 'system', which are directly relevant to SF at the 'cruciform' geometry of its ground state (see Fig. 1a and Supplementary Note 1). These states are the (anti)symmetrised local exciton states $LE^{\pm}$, the (anti)symmetrised charge-transfer states $CT^{\pm}$ and the final triplet-pair TT state. The $LE^{\pm}$ and $CT^{\pm}$ states are coherently delocalised over the two monomer units, leading to an optically bright $LE^{+}$ and dark $LE^{-}$ state (J-dimer)[37,38]. The spontaneous generation of pairs of triplet excitons results from the decay of the optically excited $LE^{+}$ bright state to the lower-energy TT states. We find excited-state energies of 2.07 eV ($LE^{+}$), 2.20 eV ($LE^{-}$), 2.75 eV ($CT^{\pm}$) and 1.83 eV (TT), which define the diagonal system Hamiltonian $H_{S}$. This makes for an ideal system for studying the role of vibronic quantum dynamics in SF, as, crucially, the molecular symmetry of the ground-state geometry suppresses all electronic couplings between these excited states. Thus, although SF is strongly exergonic in this dimer, it is strictly forbidden unless this symmetry is broken by vibrational motion (see Supplementary Figure 2). However, as in many SF systems, even symmetry breaking cannot induce a significant direct coupling between $LE^{+}$ and TT, as the matrix element for this two-electron process is almost always very small[5,6]. Instead, SF is indirectly mediated by the higher-lying CT states via superexchange[5,6], which is why they must be included in both the

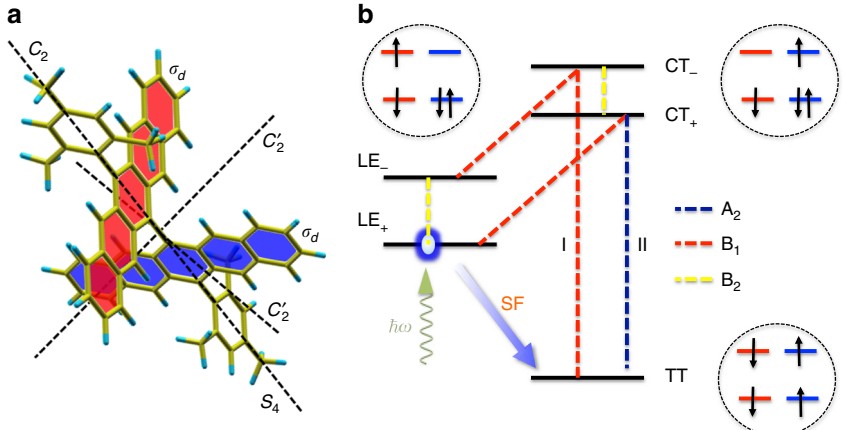

**Fig. 1** Molecular structure and electronic modelling of singlet fission in 13,13'-bis(mesityl)-6,6'-dipentacenyl. **a** 13,13'-bis(mesityl)-6,6'-dipentacenyl (DP-Mes) showing symmetry elements at the ground state geometry computed by DFT (point group $D_{2d}$). Pentacene monomers also indicated (red, blue). **b** Illustration of the ab initio electronic model showing energy-level ordering (not to scale) and typical electronic configurations defining the locally excited (LE), charge transfer (CT) and doubly excited triplet-pair (TT) states. Dashed lines indicate interstate vibronic couplings labelled by normal mode symmetries. The initial photoexcited state (LE$^+$) is not coupled to TT; it must decay via vibronic superexchange mediated by CT states. This is a second-order process, so can involve two interactions with the same environment (path I, coherent) or different environments (path 2, incoherent). $A_1$ 'tuning' modes (not shown) shift the energy levels of each state but do not induce any interstate coupling

ab initio and dynamical simulations, even if they are never populated[6]. As couplings between states arise exclusively from vibronic effects, superexchange must also be accompanied by the creation and destruction of virtual vibrational quanta in modes that link the LE, TT and CT manifolds, which we term vibronic superexchange (VSEX). Figure 1b shows some of the potential pathways and modes involved in the VSEX for DP-Mes, showing that different modes may participate in this second-order process to yield both coherent couplings (same mode mediates VSEX) and creation/emission of real quanta (two distinct modes mediate VSEX).

To explore these effects, we then employ further DFT calculations to construct a linear vibronic Hamiltonian[39]

$$\hat{H}_{\mathrm{LVC}} = H_{\mathrm{S}} + \sum_{n=1}^{252} W_n \frac{(\hat{b}_n^\dagger + \hat{b}_n)}{\sqrt{2}} + \hat{H}_{\mathrm{VIB}}, \quad (1)$$

based on 252 quantum harmonic normal modes $\hat{b}_n^\dagger$ of the dimer's ground-state potential. This describes the emerging interstate electronic couplings $W_n$, as the system is perturbed from the ground-state geometry (see Supplementary Note 1). Note that the number of modes considered the model, which is slightly smaller than the total number of modes of the dimer structure shown in Fig. 1a (318). This is due to the removal of low-frequency modes ($< 50\,\mathrm{cm}^{-1}$) for which the harmonic approximation is inappropriate, and also the exclusion of the weakly coupled, very high-frequency modes associated with the pendant side group atoms. We also found that our results showed only small quantitative changes if we reduced our model to just the (75) modes that account for 90% of the system–bath coupling in each environment cluster (see below), although this reduction does not lead to any advantages in terms of simulation speeds or memory, as shown and discussed in Supplementary Note 7.

Although for clarity we illustrate our technique with a particular example, our method is designed to deal with linear vibronic models for fairly arbitrary parameter sets. This greatly extends the domain of applications for our method, and the general form of Hamiltonian in Eq. (1) has been applied to systems as diverse as photosynthetic proteins and the strongly coupled electron–phonon–photon systems found in

semiconductor heterostructures[28,40–43]. The model also remains unsolved and an important testing ground for new theory and computational techniques[44], and we also note that its rich phenomenology has recently been the focus of experimental quantum simulations using superconducting and ion trap qubit architectures[45,46].

**Interface to TTNS.** Following the intuition developed in MPS approaches to model systems, we then perform a unitary transformation to cast $\hat{H}_{\mathrm{VIB}}$ into an equivalent system of independent 1D-chain environments that interact with the system through just one site of the chain (Fig. 1c). This orthogonal polynomial chain transformation[27,30,31] directly exposes the low-entanglement properties of the environments, i.e., their effective 1D and nearest-neighbour coupling structure, which are essential for their efficient simulation with tensor wave functions (vide infra). However, this type of transformation can only be applied to modes that, up to a scaling factor, couple to the system in essentially the same way, so that their couplings can be factorised as $H_I = W \sum \lambda_n (a_n + a_n^\dagger)$. While this is often the case in toy models or solid-state systems (allowing the entire environment to be lumped into a single 1D chain), the molecular ab initio couplings $W_n$ generally show significant $n$-dependent variations that grow with the irregularity and size of the molecular structure.

In order to reduce the computational complexity as much as possible, we introduce a clustering operation over the set of $W$ matrices. Defining a 'coupling pattern' $\overline{W}_n$ such that $W_n = \lambda_n \overline{W}_n$, we group modes with similar $\overline{W}_n$ by employing unsupervised machine learning in the form of K-means clustering[47]. This determines the minimal required set of sub-environments $i$ and their coupling operators $\overline{W}_i$ that optimally approximate $\overline{W}_n$ of assigned modes, giving the best decomposition of the system-mode coupling of the form $H_I = \sum_i \overline{W}_i \sum_n \lambda_{in} (a_{in} + a_{in}^\dagger)$, where the index $n$ runs over the total number of modes in cluster $i$. The clustering algorithm shows that the DP-Mes couplings require a minimum of four clusters, which is perhaps not so surprising; by group theory, modes should naturally break up into clusters labelled by each 1D irreducible representation (irrep) $A_1$, $A_2$, $B_1$ and $B_2$ of the

approximate point group $D_{2d}$ of the DP-Mes ground-state geometry (see Fig. 1a). This validates our more general clustering approach, suggesting the particular efficiency of clustering for systems with well-defined symmetries. However, there can also be large variances of the coupling patterns around the centroid $\overline{W}_i$ within a given symmetry class. For DP-Mes, the clustering algorithm determines that seven clusters are needed for a faithful representation of the original ab initio model, which can be seen clearly in the visualisation of the coupling clusters presented in Supplementary Figure 3. To the best of our knowledge, K-means clustering has never before been applied to open quantum system problems, and exploring the formal aspects of this type of analysis may generate interesting new insights into efficient representations of system–bath Hamiltonians. When referring to individual cluster environments, we use the symmetry label first, then an additional index to label each subdivision within the symmetry class. For example, the two sub-clusters of $A_1$ symmetry modes are denoted as $A_{1,1}$ and $A_{12}$.

The transformation of each cluster into a chain $\hat{H}_{c,i}$ can now be performed, yielding the star-like Hamiltonian (see Fig. 1c)

$$\hat{H}_{\text{Star}} = H_S + \sum_{i=1}^{7}\left[\overline{W}_i\|\lambda_i\|\frac{(\hat{a}_{i,0}^\dagger + \hat{a}_{i,0})}{\sqrt{2}} + \hat{H}_{c,i}\right], \qquad (2)$$

where $\lambda_i = (\lambda_{i_1}, \ldots, \lambda_{i_n})$, and $a_{i,0}^\dagger$ is termed the reaction coordinate (RC) of chain $i$, which directly reflects collective motion and time scales of each independent environment. The matrices $\overline{W}_i$ and parameters of our custer model are given in the Supplementary Note 2, and an overview of the processing steps is shown in Fig. 2.

**Tensor networks for vibronic quantum dynamics.** The critical challenge for simulating the complete wave function dynamics of Eq. 2 is its sheer size: each vibrational mode must be described by a Fock state basis of $n_b > 100$ states, leading to a many-body state formally described by $\approx n_b^{252}$ amplitudes. This is a generic problem, the so-called curse of dimensionality. However, for the short-range Hamiltonian of Eq. (2), most of this exponentially large Hilbert space is never explored[48]. This realisation motivates us to apply a tree tensor network state (TTNS) ansatz[49,50], which is a powerful low-rank tensor approximation scheme that decomposes the wave function to yield a connected network of small, individually updatable tensors, as sketched in Fig. 3a. We note that very important parallel develoments in the theory of low-rank approximations and tensor decompositions have also been made in the applied mathematics community, with work on Hierarchical Tucker formats and Tensor Trains (formally equivalent to MPS) being of particular relevance for the representation of large quantum states[25,26,51,52]. Full and formal details of our wave function decomposition are given in Supplementary Note 3, and references within, so here we shall just give a sketch of the concept.

The exponentially large tensor (multidimensional array) $C$ that stores all of the configuration amplitudes is decomposed into a product of smaller tensors, one for each of the vibrational modes and the electronic system. The ordering of the product of tensors normally follows the topology of the underlying Hamiltonian, and is easy to define for 1D nearest-neighbour systems. The Hamiltonian we simulate is a collection of chains joined at one site, hence the ordering of our tensors takes a branching tree-like form (Fig. 3). For the vibrational modes, the tensors take the form $A_{ijk}$ where the k index runs over the $n_b$ basis states (the 'physical' index). For a specified physical state (fixed k), the two other indices (dimensions) describe a matrix (as in matrix product states[21,22,29–31,33,53]), and the contraction over these indices when

the matrices of neighbouring vibrational modes are multiplied together yields the amplitude of a configuration (Fig. 3a). Thus, for a fixed bond dimension that is sufficiently large to capture the entanglement appearing over the dynamics, this type of ansatz effectively creates a linear-scaling problem in system size. However, this becomes a higher-order polynomial when including the overheads for tensor contractions, the sequential updating of each tensor and the extraction of observable quantities when simulating the dynamics[21–23].

A central parameter is the size of these matrices, quantified by the 'bond dimension' $D$. The degree of correlation that can be described by a tensor network ansatz is set by $D$[21–23], with $D = 1$ corresponding to a simple product state (mean field) ansatz. We find that $D$ values in the range of 10–100 are needed for converged results, indicating the entangled nature of the dynamics of our model. The chain tensors $A$ are relatively 'cheap' to simulate (each is of size $n_b D^2$), which is the essential motivation for the chain transformation. However, the central tensor which represents the electronic system (S in Fig. 3a), suffers a curse of dimensionality, as it is connected to all of the chains and scales as $D^7$ in our fission example. Although clustering and chain transformations reduce its $N$-neighbours from 252 in $\hat{H}_{\text{LVC}}$ down to 7 in $\hat{H}_{\text{Star}}$, the central tensor still requires 57 TB of memory.

Remarkably, we can compress the tensor down to 40 MB by further decomposing it into a loop-free tree network of smaller auxiliary entanglement renormalisation (ER) tensors[23,54] (see Fig. 3b). Unlike the tensors associated to the physical vibrations and electronic system, these ER tensors are a decomposition of the system tensor purely in terms of the correlations that exist between it and the environments. Consequently, they do not carry physical indices, and the number and connectivity of these objects is chosen to exploit inter-environment entanglement (blue lines in Fig. 3) to minimise the von Neumann entropy $S = -\text{Tr}(\rho \ln \rho)$ across all bonds. In the example shown in Fig. 3b, the sum of the entropies of the reduced states of chain $A_{11}$ and $A_{12}$ is seen to be greater than the entropy of their joint reduced state, suggesting that they act on the system in a correlated or collective way. This further suggests that we can apply a type of isometry (renormalisation), which is implemented by connecting them together by an auxillary tensor that then transfers a reduced number of effective correlated states into the network where they will eventually act on the system. This general procedure maximises the achievable accuracy at fixed $D$, reducing the total complexity of any tensor in the network to $O(D^3)$. In our SF example, this decreases the amount of memory required by six orders of magnitude, which is the key enabler for applying TTNS to $N > 3$ environment models. We note that while entanglement renormalisation methods have been proposed before in condensed matter settings[23,54], the open system problem appears to be particularly well-suited for their application. This is due to the fact that very strong correlations are 'concentrated' around the electronic system and can be decomposed with just a few layers of ER tensors after which the systems can be described with simple A-type matrix tensors for the independent (non-interacting) chains. Full details and further discussion are given in Supplementary Note 3. Finally, our TTNS is time-evolved with a recent time-dependent variational principle (TDVP) that has been especially adapted to handle the topology and mixed tensor composition (physical, ER, isometries) of our wave function[31,36,55,56]. TDVP has proven to be faster than competing algorithms[31,56,57], and allows simulations to run for longer times without rapid growth of bond dimensions[58]. The essential feature of this approach is that each tensor is sequentially time-evolved with a local effective Hamiltonian $H^{\text{eff}}(t)$ that is constructed at

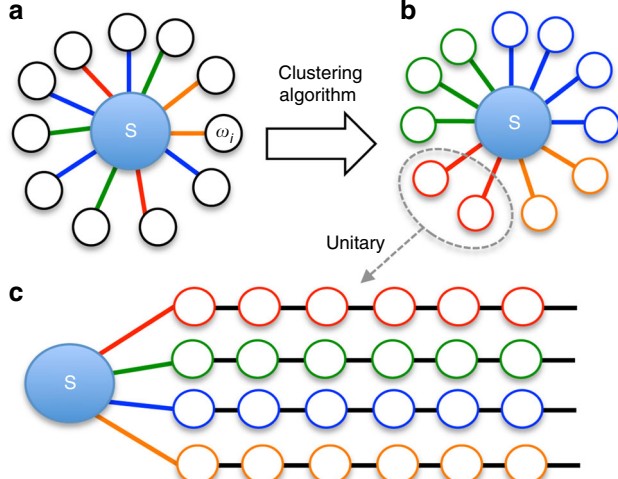

**Fig. 2** Ab intio information processing and model extraction. **a** The ab initio linear vibronic Hamiltonian describes how each vibration mode of frequency $\omega_i$ (outer circles) couples to the system (S). The coupling matrix for each mode could be different, shown as the colour of connecting lines. **b** A K-means learning (clustering) algorithm groups modes according to their common coupling type (colour grouping in figure). **c** A unitary transform (linear combination of vibration modes) of each cluster creates an equivalent nearest-neighbour chain representation of the Hamiltonian, ideal for tensor network simulation

each time step from the full many-body state (see Supplementary Note 3). This is not only a very efficient numerical strategy for evolving the state; we shall also show that the $H_{\text{eff}}(t)$ acting on the system tensor also contains valuable information that can be used to visualise multidimensional dynamics.

We now present numerical results for singlet fission in DP-Mes. All simulations are performed at zero temperature, with no environmental excitations, and start initially from the optically bright $LE^+$.

**Fission dynamics in DP-Mes**. Figure 4 presents numerically exact electronic diabatic population dynamics (a) and the environmental occupation of the RCs across different orders of magnitude (b, c). Vibrationally mediated SF happens on a time scale of 200 fs (Fig. 4c), with a final TT yield of ≈90%, comparable with experiments where yields around 94% and time scales of 400–700 fs have been reported[37]. Considering typical instrument response times (100 fs), the limitations of TD-DFT in vacuum and the fact that experiments were performed in solutions, this result appears entirely reasonable. The dynamics are multiscale; at early times < 20 fs they are dominated by ultrafast damped Rabi oscillations ($LE^+ B_1 CT^+$, $LE^+ B_2 LE^-$, see Fig. 4a), with significant population transfer between $LE^+$ and $LE^-$ accompanied by rapid oscillatory excitation of the RCs of the $B_2$ environments that couples these states (see Supplementary Figure 10). By Rabi oscillation, we make the analogy to the Rabi model of a quantum photon mode that drives an atomic transition between two levels, with the quantised molecular vibrations playing the role of photon. The oscillations arise from reversible exchange of quanta between the transition and field, effectively resulting in the emergence of entangled vibronic (dressed) states. From another point of view, what is being resolved here in the diabatic basis could also be seen as the real-time formation and early motion of the adiabatic states from $LE^+$ along the coupling-mode coordinates of $B_1$ and $B_2$. The entanglement generated by the formation of these states can also be seen in rapid (<20 fs) rise of the von Neumann entropy of the

reduced electronic state (see Supplementary Figure 9). This is interesting, as the time scales of this evolution just fall within the time resolution of ultrafast spectroscopies, suggesting these non Born–Oppenheimer dynamics might indeed be observable.

The Rabi oscillations of $CT^+$ also correlate with the initial increase of TT population $d\rho_{TT}TT/dt$ indicating a small coherent mixing of TT character into the photoexcited state via super-exchange path I mediated by the $A_2$ modes (Fig. 5). However, the more important mixing is that with $LE^-$, which we now show opens the dominant superexchange pathway (2) that drives efficient fission from the initial photoexcitation[12,59]. After > 50 fs, the population of the TT state quickly begins to rise, mirroring the decay of the hybridised $LE^+$ and $LE^-$ states, and accompanied by strong excitation of the $B_1$ RCs. The $B_1$ vibrations only couple $LE^-$, TT and $CT^-$, and thus the mixing of $LE^-$ character into the initial state opens the vibronic superexchange channel, labelled path 2 in Fig. 1. As expected from super-exchange, the $CT^-$ state acts only as a virtual mediator for the effective vibronic coupling between $LE^+$ and TT; it is never appreciably populated, as seen in Fig. 4a. The appearance of a strong imaginary coherence Im ($\rho_{TT,LE^-}$) in the reduced density matrix of the electronic system indicates coherent transport between these states, which is only possible because the same environment ($B_1$) mediates both couplings that lead to the superexchange interaction (no real vibrational quanta are emitted in the process). Figure 5 shows that the dynamics of this coherence also matches the population current in the TT state $d\rho_{TT}/dt$, providing strong evidence that the $B_1$ superexchange drives SF in DP-Mes.

The spontaneously generated coherence between TT and $LE^-$ is the result of a higher-order vibronic interaction (super-exchange), and might give a characteristic signature in two-dimensional Fourier transform spectroscopies[12]. We further note that after a rapid rise in entropy, the formation of TT does not purify the reduced electronic state, i.e., the electronic and environmental degrees of freedom are entangled (cannot be factorised). This cannot be explained by a simple relaxation process from a high-energy to low-energy state with the incoherent emission of vibrational quanta (at $T = 0K$); we shall indeed demonstrate that the ultrafast coherent mechanism described above can also be seen as a type of adiabatic avoided crossing phenomenon.

Finally in this section, we note the following observation of relevance for future experiments. The residual population (≈10%) in the higher-lying diabatic states shows that we reach a bound TT state which is lower in energy than a T+T (or non-relaxed TT state) at long times, and has a weakly mixed character[60–62]. In particular, the non-vanishing $LE^±$ character indicates that the equilibrium 'TT' state can exhibit singlet character in its excited-state absorption[37], and the TT population is modulated over the dynamics by the fast $A_{1,2}$ tuning modes, which could be observable in ultrafast vibrational spectroscopy[12,13].

**Dynamic energy surfaces**. The preceding section demonstrated how we could use standard observables of both the system and environment to unravel the superexchange pathway and various modes that contribute to fission. Now, we introduce another powerful way of visualing and analysing the dynamics based on energy surfaces. We do this by calculating dynamical adiabatic potential energy surfaces (PES) and also total energy surfaces (TES), which include the non-adiabatic nuclear kinetic energy operator. A major difficulty of visualising energy surfaces for N > 2 vibrational modes is the multidimensional nature of the dynamics; trying to visualise dynamics—where possible—requires physical intuition, or some other prior knowledge, to identify a sufficiently small number of 'reaction coordinate' modes that

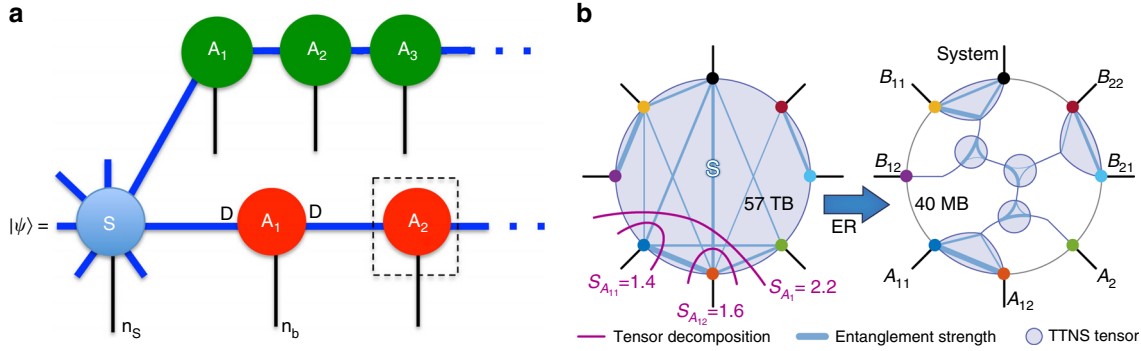

**Fig. 3** Tensor network state construction and entanglement renormalisation analysis. **a** The many-body wave function is decomposed into a system of individual tensor objects associated to each environment mode (A) or the system (S). Black lines represent a physical tensor index running over the possible states of the system, i.e., $n_b$ Fock states for vibrations. For a given physical state, an A tensor is a matrix of dimension $D^2$, and the amplitude of a complete configuration of the wave function is given by the contraction (matrix multiplication) indicated in the figure (only two environments are shown for simplicity). The tensor for the system is 8th-order, due to its connections to seven environments and scales as $D^7$ in bond dimension. **b** By analysing the entanglement entropy of different system–bath partitions, it is possible to decompose the system tensor into a network of third-order auxiliary tensors (without physical indices), reducing the local connectivity in the network in an optimised way. Discrepancies in entropy $S$ between multiple tensor decompositions (purple) reveal the tensor's entanglement structure (blue) and guide the ER-network design (see main text)

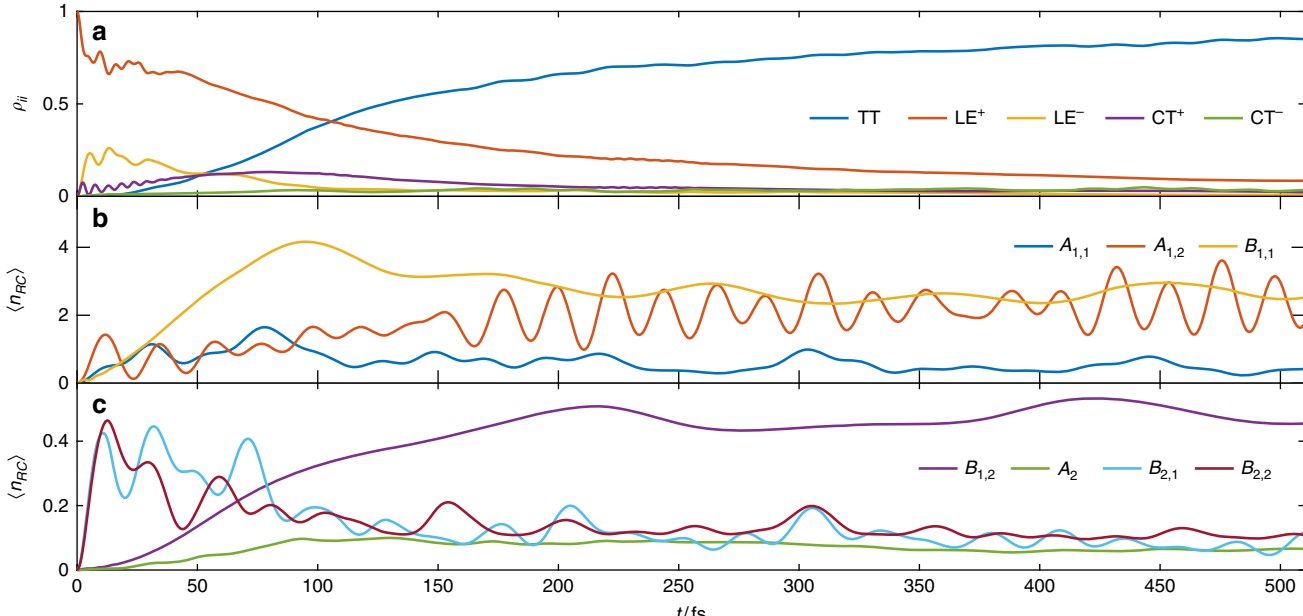

**Fig. 4** Simulated SF dynamics in DP-Mes. Electronic populations (**a**) and environmental RC occupation dynamics (**b**, **c**). For <20 fs, damped Rabi oscillations transfer the population from $LE^+$ to $LE^-$ and $CT^+$. At subsequent times TT mirrors the $LE^+$ decay leading to SF on a time scale of 200 fs and suggesting a direct SF pathway. Similar time scales are present in the bath excitation $n_{RC}$ of $B_1$ and $B_2$. The final TT yield reaches 90% after 1 ps

are relevant for the process under study. Here, we exploit the fact that our pure wave function can always be Schmidt-decomposed into a system–bath bipartition of the form $|\psi(t)\rangle = \sum_{i=1}^{5} \lambda_i |i_S(t)\rangle |\phi_i(t)\rangle$, where the system states $|i_S(t)\rangle$ and many-body environmental states $|\phi_i(t)\rangle$ are mutually orthogonal in the separate system and environment Hilbert spaces and $\lambda_i(t)^2$ is the probability of finding the environment in the $i$th state at time $t$. Obtaining this decomposition is particularly easy in a tensor network ansatz, and follows from applying a singular value decomposition (SVD) of the system tensor. The SVD is actually used throughout the algorithm to update and normalise the state, so access to this information is essentially 'free'. We then calculate the energy surfaces at each time point as the eigenvalues of the effective Hamiltonian $H^{\text{eff}}(t)$ and potential $V^{\text{eff}}(t)$ of the system states generated by the bath

configurations $|\phi_i(t)\rangle$

$$H^{\text{eff}}_{(ij),(k,l)}(t) = \langle \phi_i(t) | \langle j | \hat{H} | \phi_k(t) \rangle | l \rangle, \tag{3}$$

$$V^{\text{eff}}_{(ij),(kl)}(t) = \langle \phi_i(t) | \langle j | \hat{H} - \hat{T} | \phi_k(t) \rangle | l \rangle, \tag{4}$$

where $i$, $k$ are index bath states and $j$, $l$ denote the electronic states. In our SF example, the effective Hamiltonian $H^{\text{eff}}(t) \in \mathbb{C}^{25 \times 25}$ leads to 25 surfaces (five electronic states lead to five distinct environment configurations). However, it is important to note that these surfaces are effectively an energy spectrum parameterised just by time and therefore easy to visualise. The exact dynamics provide us with the dominant environment configurations at each point in time, enabling us to simply follow these

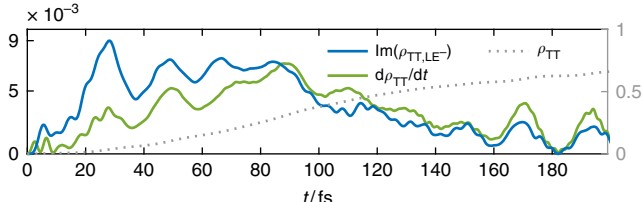

**Fig. 5** Evidence for coherent transport. The change $d\rho_{TT}/dt$ of $\rho_{TT}$ correlates with the dynamic coherence $Im(\rho_{TT,LE^-})$ indicating coherent superexchange via $B_1$. The large oscillations with period of 20 fs correlate with $A_{1,2}$ oscillations (see Fig. 4b). The maximum SF rate is achieved at 90 fs during the passage of the avoided crossing in the TES shown in Fig. 6b

motions without reference to any paticular molecular coordinate or normal mode basis. With some further manipulation (see Supplementary Note 5), we can now look into the real-time energetics of the dynamics (Fig. 6a, b) by showing the occupations and diabatic character of the excitations on these surfaces. In Figure 6a, b, we represent energy surfaces as lines, the overall population of each surface by the width of a fill around the line and the diabatic composition of these occupations by the colours of the fills.

Returning to singlet fission, we see that the dynamics in both the PES and TES are confined to approximately two low-energy surfaces. The SF dynamics in the adiabatic PES (Fig. 6a) are evident as the population transfers from the lowest $LE^+$ surface to the lowest TT surface. Since the population transfers between non-crossing surfaces, this displays the non-adiabatic nature of the dynamics. Additionally, the adiabatic PES of the populated TT (blue) stays consistently below the $LE^+$ PES (red, Fig. 6a), thus SF in DP-Mes should not be caused by a conical intersection as proposed for other SF systems[13,15]. However, the actual sequence of processes leading to fission is best resolved in the non-adiabatic TES (Fig. 6b). Here, SF occurs by an avoided crossing appearing at 80 fs (black arrow) which is caused by three key 'events': (1) the unoccupied TT TES is pushed above the $LE^+$ TES at 20 fs (red arrow) by $A_{12}$ and $B_{22}$ motions; (2) the TT is increasingly coupled to the $LE^+$ by superexchange caused by growing $B_1$ displacements; (3), the same $B_1$ motion causes level repulsion with the high-lying CT states (see Fig. 6a) and a slow descent of the TT surface towards that of the occupied $LE^+$ state. The combination of (2) and (3) results in an avoided crossing with a splitting of ≈11 meV at 80 fs, in which the population on the $LE^+$ surface evolves adiabatically (in the Landau–Zener sense) into a 95% TT population. Indeed, estimates based on the Landau–Zener formula for the diabatic transition probability predict a coupling of 10 meV, which is encouragingly close to our result (see Supplementary Note 10).

This last picture can be further complemented by looking at the dynamical time scales associated with the reaction coordinates mediating these events: in (1), the energy increase of the (unoccupied) TT TES at 20 fs is caused by the response of the collective modes $\omega_{RC,A_{12}} \approx 1377\,cm^{-1} \sim 24\,fs$ and $\omega_{RC,B_2} \approx 1100\,cm^{-1} \sim 30\,fs$ to the sudden occupation of the $LE^+$ state; in (2), the effective vibronic coupling emerges from displacement of $B_1$ with $\omega_{B_1} \approx 400\,cm^{-1} \sim 83\,fs$, setting the longer time scale for SF (Fig. 4a–c); in (3), the slow reorganisation of $B_1$ increases the effective TT-$LE^+$ coupling from 3 meV at 20 fs to 6 meV at 80 fs, as seen from the avoided crossing gaps. We therefore see that no population is transfered when the TT surface initially rises through the $LE^+$ surface, as this is caused by $A_1$ mode displacements, which respond much faster than the $B_1$ modes.

Consequently, the superexchange coupling between surfaces is small at the first crossing (small $B_1$ displacment) and transfers little population due to the very fast crossing (non-adiabatic, in the L-Z sense). However, the initial motion of the TT surface above the $LE^+$ surface is crucial for the subsequent descent to the avoided crossing, so both $A_1$ and $B_1$ modes play important, albeit different, roles in SF in DP-Mes.

**Remarks on simulation performance**. For the fission results presented above, convergence was obtained with 100 bosonic Fock states per vibrational mode. The dynamics of all observables converged to ≈1% for a maximum TTNS bond dimension $D$ between ER-nodes of $D_{Node,max} = 80$, while along the chains $D_{chain,max} = 30$ (see Supplementary Figure 5). Thus we cover a Hilbert space of $10^{500}$ states using only $10^7$ parameters. Moreover, the typical simulation times for these parameters were ~96 h on a single Intel Xeon E5-2670 core for a trajectory of a picosecond, which also included the computation of all observables and the energy surface visualisations that have not been previously reported in other methods. A complete set of convergence data w. r.t. bond dimensions and CPU times can be found in Supplementary Note 4. As a system-specific model of a particular molecule, we are not able to compare this numerical performance with another method directly, but on the basis of published results on the spin-boson model (SBM)[19,20,28,30,31], we believe that our algorithm should be broadly comparable with ML-MCTDH in terms of resources and speed, although SBM simulations with over 10 K modes have been demonstrated with ML-MCTDH. However, we note that the procedure of creating 'balanced trees' for efficient ML-MCTDH calculations appears, for the moment, to be a matter of trial and error, as discussed at length in the context of singlet fission by Zheng et al.[11]. Generically, efficient tree constructions are discussed in greater depth in a recent review by Wang[20]. Our entanglement (von Neumann entropy) analysis of the core tensor and the construction of an optimal three-legged network using ER nodes is based on a clear mathematical framework and might also be applicable to ML-MCTDH problems. Compared with ML-MCTDH and the present method, the recent MPO-based TEMPO algorithm of Strathearn et al. appears to require slightly longer computational time and more computational resources for the Ohmic spin-boson model at zero temperature and $\alpha := 0.5$[34], as do techniques based on hierarchical equations of motion[35]. However, these latter methods are also efficient in treating the effects of finite temperatures and can reach long times without recurrences, which make them very attractive for looking at experimentally relevant reduced system dynamics.

We finally remark that we have also explored convergence w.r. t. the numbers of modes retained in the clusters (see Supplementary Note 4). We found that a reduction of the number of modes to just 75 still gave quantitatively similar dynamics, although the computational time and resources needed for the simulation did not greatly decrease. We previously mentioned that each 1D environment is relatively 'cheap' to extend, but we believe that the ability to shrink the environments dramatically is related to the dominance of corrrelations between the system, the RC modes and a relatively small number of nearby oscillators in each chain on the physics. Indeed, we have also explored the effect of deliberately altering the parameters of the chains to yield RC modes that are damped by a simpler, extended 'outer' environment (490 modes). This leads to fairly good qualitative agreement with the full *ab* initio model, and could provide an interesting way to benchmark a range of theories based on simpler RC master equations[63–65].

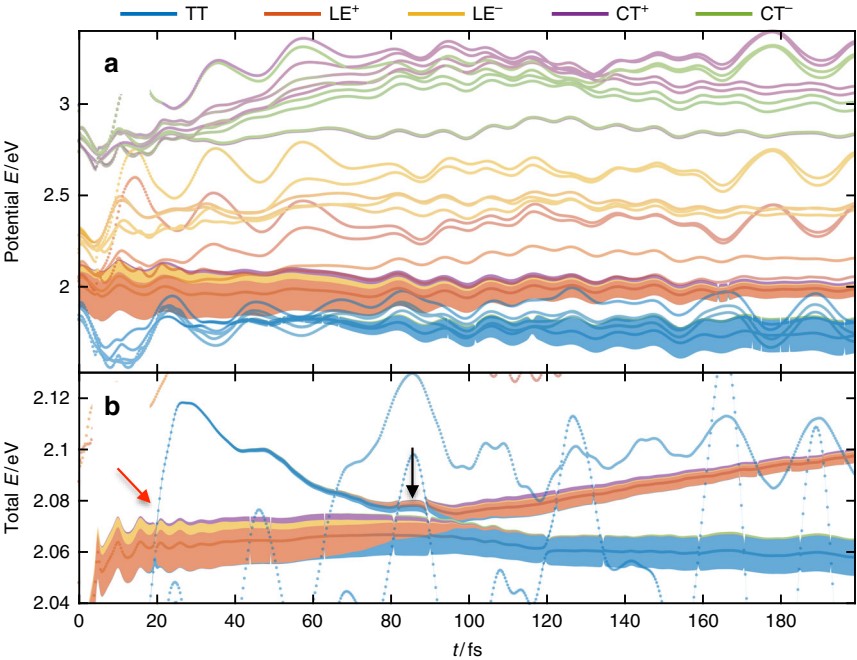

**Fig. 6** Energy surfaces and their electronic populations. Each bath state $\phi_i$ ($i = 1$–5) creates its own set of five energy surfaces, computed as the eigenstates of $V_{eff}(t)$ (PES) shown in (**a**) or $H_{eff}(t)$ (TES) shown in (**b**). The filled areas indicate the amount of the electronic population on each surface. The surfaces are coloured according to their amplitudes of each diabatic electronic states they contain. The red and black arrows on the TES indicate important events that lead to fission by an adiabatic avoided crossing, as decribed in the text

## Discussion

The results above demonstrate that many-body tensor network methods and entanglement analysis can provide a powerful bridge between ab initio electronic struture and real-time open quantum dynamics, and provide a truly microscopic elucidation of mechanisms underlying ultrafast light-harvesting processes. Moreover, as a general computational strategy, our technique should also be applicable to a wide range of complex non-Markovian systems in physics and chemistry, and may allow similar insights into timely problems where the boundary between systems and environments may become a subtle point, such as few-body quantum thermodynamics and non-equilibirum phenomena in many-body systems. Indeed, from an open system perspective, the coordinated sequences of processes seen in SF— each involving different types of environment and the emergence of different types of interactions between the electronic states—is a good example of non-Markovianity in action.

For light-harvesting systems, a number of further developments also appear possible at this point, including the possibility of reversing several of the transformations we perform to extract information about the original molecular normal modes that drive the dynamics. With this information, physical reaction coordinates could be identified and targeted in future experiments. An intriguing idea is that light-harvesting processes might then be controlled via these motions, either through quantum control protocols or strong light–matter (polaritonic) interactions[66,67]. In both cases, access to detailed and accurate dynamics is essential, and we further note that—as the single-core computational costs of our technique allow easy parallelisation— potentially high-throughput simulations could be implemented for dynamics-based screening of such SF materials. We therefore hope that the approach we have set out could provide a useful tool in the emerging drive towards dissipative coherent devices constructed from rationally designed, nanostructured organic materials[1–4].

## Methods

**DFT**. All DFT calculations were performed with the NWChem electronic structure code in vacuum[68]. The ground-state structure and vibrational modes were obtained employing analytical Hessians at the cc-PVDZ/B3LYP level of theory. Excited-state energies and forces (corresponding to the diagonal elements of $W$) for LE$^\pm$ and CT$^\pm$ were calculated from (linear-response) TDDFT gradients at the cc-PVDZ/LC-BLYP level of theory. The long-range corrected functional is required to correctly describe the $S_1$ state of pentacene as well as the states with charge-transfer character. An optimised range-separation parameter $\mu = 0.29$ was used in the LC-BLYP functional. This choice gives a good description of the energy of the first excited singlet of the pentacene molecule[69]. For TT, the quintet state (total spin 2) was used as a proxy for the purpose of calculating energy and forces. The frontier molecular orbitals at the cc-PVDZ/LC-BLYP level were also used as inputs for the evaluation of the off-diagonal couplings. Modes below 110 cm$^{-1}$ and above 1500 cm$^{-1}$ were disregarded due to either unrealistic ab initio parameters arising from the neglect of their anharmonicities or irrelevance for present time-resolved experiments, respectively. Limitations of the method are discussed in the Supplementary Note 1.

**Tensor network states**. To prepare the linear vibronic DP-Mes Hamiltonian for the TTNS simulations, the vibrational modes were clustered to independent environments by using a weighted K-means algorithm[47]. These environments were then mapped onto chain Hamiltonians with the orthogonal polynomials transformation to facilitate the tensor tree network decomposition and numerical evolution of the complete vibronic wave function[27].

The model was simulated with a TTNS, as depicted in Supplementary Figure 4, incorporating entanglement renormalisation (ER) tensors connecting the vibrational chain environments to the excitonic states and using an optimised boson basis (OBB) to allow for a large and expandable Fock basis. The time-evolution is performed in the time-dependent variational principle (TDVP)[36] to simulate exciton–phonon dynamics at zero temperature with no initial environmental excitations at a time step of 0.33 fs. Details about the TDVP algorithm we have developed for the TTNS evolution are given in the Supplementary Note 3, and further background can be found in refs. [31,36,49,50,55]. Energy surfaces were calculated from the effective Hamiltonian and effective potential of the system tensor. Further computational details are provided in Supplementary Notes 3, 4 and 5.

## Code availability

The custom tree tensor networks state code used for simulations is available by written request to the corresponding author.

## Data availability

The data underlying this publication are available in the University of Cambridge data repository at www.repository.cam.ac.uk.

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

## Acknowledgements

We thank R.H. Friend for making this work possible and A. Rao for useful discussions. F.A. Y.N.S., D.H.P.T., N.D.M.H. and A.W.C. gratefully acknowledge the support of the Winton Programme for the Physics of Sustainability and the Engineering and Physical Sciences Research Council (EPSRC). A. J. M. acknowledges support from EPSRC grant EP/M025330/1.

## Author contributions

F.A.Y.N.S. developed the TTNS method and performed calculations and D.H.P.T. performed DFT calculations. F.A.Y.N.S., D.H.P.T. and A.W.C. interpreted results and wrote the paper. A.J.M. initiated the project and provided experimental insight into DP-Mes. A. J.M., N.D.M.H. and A.W.C. supervised the project. All authors discussed results and contributed to the paper.

## Additional information

**Competing interests:** The authors declare no competing interests.

