## [Peer Review File · Nature Communications]

Reviewers' comments:

Reviewer #1 (Remarks to the Author):

This paper presents a detailed and very thorough numerical model of singlet fission in the organic molecule, in which vibronic effects play a key role.

The computations rely on a rewriting the Hamiltonian of the electronic and vibrational level of the system as a tensor network. There are two main elements. First, the 252 couplings between different vibrational levels and the 5-level electronic system are grouped into different "families" which interact roughly through the same system operator. Each family can then be represented as a tensor chain coupled back to the system. Second, the resulting central system "star" tensor is condensed through an entanglement renormalization tensor. That this level of calculation is possible at all is extremely impressive - and represents an intellectual tour de force. It also, importantly, gets the singlet fission rates roughly correct, and is able to provide insight into the mechanism by which this occurs. The calculation contains several new elements - the grouping of the vibronic elements into families in an optimal way, and the re-expression of the system tensor, and the combination of these is clearly very powerful and could be applied to other systems I think. I'm therefore broadly in favour of moving this work towards publication, though I do have several comments and suggestions which the authors must address before a final recommendation.

More major:

1. Comparison to other methods. On Page 1, the authors say that, for example, MCTDHF, can be used for this kind of problem. I would like to see the authors return to this in their conclusion. How does the method here compare to other methods in terms of speed and accuracy? How do the authors know their method is accurate?
2. Can the authors make a comment about how easy it would be to make predictions/create designs for optimised singlet fission in other materials using this method?

Other comments:

1. There are a lot of typos and similar small errors in the paper - the authors should do a careful proof read and spell check before re-submission. (A couple of examples - Fig 1 caption mediated not medaited; p4 first paragraph: dimensionality not dimensionally.)
2. Top of P2, cite further work on using MPS to model open systems e.g. recent paper arXiv:1711.09641.
3. Fig 1 colour scheme. Red and orange hard to distinguish for the colour blind reader.
4. P2, col. 2: why is the cruciform geometry novel?
5. P3, col 1: why are 252 modes used?
6. P3, col 1, paragraph after Eq 1. Can more references to example relevant systems be put after "linear vibronic Hamiltonian is applicable to a very wide range of molecular and supramolecular chromphoric systems,"
7. P3 col 1, next para. I think it should be Fig 2c, not 1c.
8. P3, col 2. Can the authors say in the main text how they tell that a minimum of four clusters are needed? And then what determines why 7 are needed for a "faithful" representation?

9. P3, col 2. The authors associate the clusters with the group irreps - but in Fig 1 A_1 doesn't appear. Why is this?

10. P3, before Eq 2. 1c to 2c again?

11. P4, col 1. How do the authors know that $n_b > 100$? There is also a missing reference in this paragraph.

11. P4. In discussing the linear scaling of MPS, I think it is important to say that this relies on a constant/ reasonably bounded D , obtained using SVDs and subsequent truncation.

12. Fig 6. Why are labels d and e ?

Reviewer #2 (Remarks to the Author):

I tend to agree with the editor's previous assessment. While the work is interesting and numerically sophisticated, it does not provide a clear-cut advance to justify its publication in Nature Communications. In particular, I do not see that the numerical approach described in the manuscript is completely new. Similar vibronic models have been studied extensively in the chemical physics community. The authors cited three such references and called the approach MCTDHF. None of these references were about MCTDHF, and in fact MCTDHF is a very limited method. The appropriate approach would be MCTDH and its multi-layer (ML) generalization, the ML-MCTDH method. The authors did not seem to be aware of this very relevant approach. Setting this aside, there is plenty of applied math literature out there discussing the subject, for example, SIAM J. MATRIX ANAL. APPL. Vol. 31, No. 4, pp. 2029–2054 (2010), in particular Section 5 on the relations among names like hierarchical low rank tensor format, tensor train and tree tensor format, MCTDH, sequential unfolding SVD, etc. The format of tensor trains and matrix product states (MPS) is the same. It can be viewed as a particular case of a hierarchical Tucker tensor with a comb-like tree, where each node has only at most two children, but only one of the two children has further children. The relation between MPS and the above approaches, as well as its time-dependent variations, can be found from, e.g., SIAM J. MATRIX ANAL. APPL. Vol. 34, No. 2, pp. 470–494 (2013).

Reviewer #3 (Remarks to the Author):

The present very impressive and excited paper offers a novel approach for simulating open Quantum dynamics, and provide tools for simulating light harvesting processes. The authors proposed a Tensor network discretization of an open quantum system for

Due to my own expertise I want focus on the numerical treatment.

I cannot make clear statement about the modeling. However the modeling seems to be appropriate, using the Time dependent DFT calculations to set up the the underlying operator, in particular the coupling terms. Resulting in a system bath formulation, where the bath system is generated by by (coarse grained) by dft models.

The resulting problem is a timed-dependent multi particle Schrödinger equation in second quantized form.

The author used tree tensor network states for its representation

(which is a good idea at this point) and an exponential representation of the time evolution operator, based on work of Verstaete, Lubich et al. These are advanced techniques and states of art. For the present purpose the authors have not only used best choices of methods, but they improved these methods further.

One technical difficulty with tree tensor networks is the optimization of the graph topology. In his respect the authors spend certain effort to find a good topology, which is a cornerstone for the successful realization.

The paper proposes a powerful novel approach for an area, where one needs numerical tool to simulate the (quantum) physics correctly, and which is of technological interest in photo physics

The paper applies and improves cutting edge techniques for data-sparse approximation of quantum states.

Therefore I am recommending the paper for publication, after a minor revisions according to the comment below.

Comments, corrections and minor changes.

I do not agree with the comment ... MPS are only efficient for 1D systems.....

Of course due to linear topology 1 D systems with nearest neighbor interactions are optimal for MPS.

I suggest to modify the text MPS are mostly efficient for 1D systems.....

The bibliography w.r.t to tensor networks is almost appropriate. I would like to add the monograph of W. Hackbusch

The HT tensors are tree tensor networks, However the motivation by sub-space approximation is a novelty introduced by Hackbusch, although Vidal's decomposition has been known. This perspective explains the approximation behavior of of tree tensor networks. I really appreciate the citation of Thoss & Wang, the community in MCTDH (D. Mayer et al.) has also developed tree tensor networks.

I have not looked for typos; but i have seen at least one:

page 6 column 2 line 14- (from below) : natuer..... correct: ... nature
perhaps it si, likely that there are more, Please correct typos.

The paper is dressed to experts in physics. The various definitions are hard to understand form an abstract point of view. To get a wider audience it seems to helpful to put more effort on the mathematically clear and precise definitions.

Response to Referee Reports:

Reviewer One

We thank the reviewer for their detailed report and highly constructive criticism. We were particularly delighted to see that the novel elements of our approach to simulating open systems with tensor networks (grouping of vibronic elements, entanglement renormalisation...) were appreciated by the reviewer, and we now response to his or her specific points:

1. *“Comparison to other methods. On Page 1, the authors say that, for example, MCTDHF, can be used for this kind of problem. I would like to see the authors return to this in their conclusion. How does the method here compare to other methods in terms of speed and accuracy? How do the authors know their method is accurate?”*

The use of ML-MCTDH for linear vibronic models appears to give access to similar information with similar resources, although several steps that are used in ML-MCTDH have not been formalised and incur problem-specific costs. An example - nicely discussed by Zheng *et al.* in [13] - is the building of “balanced” tree networks for the ML-MCTDH iterative hierarchical equations. This seems to be something that is largely done by trial and error, whereas we show how this can be assessed directly in tensor network methods through entanglement analysis (which we have introduced in the present work). The ML-MCTDH method also does not require chain-like structures, in principle, but will require a prohibitively large number of layers to converge, unless there is an underlying low-rank approximation that is possible (which we explicitly reveal through the construction of our transformed Hamiltonians and tensor networks). However, a rigorous comparison of our approach and ML-MCTDH on the same system or Hamiltonian remains to be performed, and we hope to that this paper may encourage such an undertaking in the near future.

In terms of accuracy, a section in our supplementary materials (SI IV) presents the methodology and results of our convergence and optimisation tests. In brief summary, a *single* core of an Intel Xeon E5-2670 on a conventional laptop produced observables converged to <3% in 4 days, and we note that our results include a large number of many body observables and simulation data that are new or non-standard, such as our on-the-fly energy surfaces (which have not, to the best of our knowledge, ever been computed in ML-MCTDH). The accuracy of the simulations can be further improved with more computational time and higher bond dimensions, as shown in the SI. We also note that the optimisation analysis shows that the most important - and demanding - resource when attempting to leverage speed and accuracy is the number of bond dimensions used to describe the interactions between the central system and the seven environments that we consider.. This further demonstrates that the entanglement renormalisation approach that we introduce is a critical component for this type of open system simulation. The same analysis also shows that the clustering and mapping of environments to 1D chains leads to very rapid convergence w.r.t the 1D chain bond dimensions, justifying the small overhead used to perform the machine learning analysis of the original Hamiltonian.

In comparison to some other methods, we also note that the recent TEMPO approach presented by Strathearn *et al.* can be compared to a previous work of ours on the spin-boson model [FAYN Schroeder and AW Chin, PRB, 2016]. For this simpler, one-environment model, we found that we could obtain converged results across the non-critical parameter space (in Ohmic and sub-Ohmic models) with a characteristic single sweep with local boson dimensions $n_k = 30$, bond dimension $D = 5$, $d_{\text{OBB}} =$

15, and $L = 200$ taking just 4 seconds on one core of an Intel Core i7-4790 CPU. Again, the output is full many body information, as opposed to a reduced density matrix, so this example compares favourably with the 20.5 hours of HPC cluster computation required by TEMPO for 500 data points for the Ohmic SBM at $\alpha=0.5$, which is a non-critical point. This calculation by our TDVMPS approach would take approximately $4 \times 2 \times 500 = 1$ hour on a good laptop, which is similar to the computational times reported by Wang and Thoss for ML-MCTDH (from 10 mins to a few hours on a single core, for $0 < \alpha < 0.5$). However, we also note that reduced density matrix techniques are typically most effective at high temperatures, whereas ML-MCTDH and our tensor network wave functions are strictly zero-temperature methods. This makes *both* types of approach extremely valuable for understanding present experiments and the full phenomenology of open systems.

We believe that this ability to compute accurate many body results for non-trivial models on common and relatively cheap computational platforms is another major plus point for our approach, and we have added a short section before the conclusions of our article to discuss the computational points and comparisons mentioned above.

Remarks on simulation performance

For the fission results presented above, convergence was obtained with 100 bosonic Fock states per vibrational mode. The dynamics of all observables converged to $\approx 1\%$ for a maximum TTNS bond dimension D between ER-nodes of $D_{\text{Node,max}} = 80$, while along the chains $D_{\text{Chain,max}} = 30$ (see Fig. 5). Thus we cover a Hilbert space of 10^{500} states using only 10^7 parameters. Moreover, the typical simulation times for these parameters were approximately 96 hours on a single Intel Xeon E5-2670 core for a trajectory of a picosecond, which also included the computational of all the system and environment observables and energy surface visualisations. A complete set of convergence data w.r.t bond dimensions and CPU times can be found in Section IV. As a system-specific model, we are not able to compare this numerical performance with another method directly, but on the basis of published results on the spin-boson model (Wang 2008, Prior 2010, Chin 2013, Schrodner 2016), we believe that our algorithm should be broadly comparable to ML-MCTDH in terms of resources and speed. However, we note that the procedure of creating "balanced trees" for efficient ML-MCTDH calculations appears, for the moment, to be a matter of trial and error, as discussed at length in the context of singlet fission by Zheng et al. (Zheng 2016). Our entanglement (von Neumann entropy) analysis of the core tensor and the construction of an optimal three-legged network using ER nodes is based on a clear mathematical/quantitative framework and might be applicable to ML-MCTDH problems. Compared to ML-MCTDH and the present method, the recent TEMPO algorithm of Strathearn et al. appears to require longer computational time and more computational resources for the spin boson model at zero temperature (Strathearn 2018), as do techniques based on hierarchical equations of motion (Shi 2018). However, these latter methods have a natural way to include the effects of finite temperature and reach long times without recurrences, which make them very attractive for looking at just the reduced system dynamics that is often all that is needed for comparison to experiments.

We finally remark that we have also explored convergence w.r.t. the numbers of modes retained in the clusters (see Section VI). We found that a reduction of the number of modes to just 75 still gave quantitatively similar dynamics, although the computational time and resources needed for the simulation did not greatly decrease. We previously mentioned that each $1D$ environment is relatively 'cheap' to extend, but we believe that the ability to reduce the environments here is related to the

dominance of the RC modes and a relatively number of nearby oscillators in each chain on the physics. Indeed, we have also explored the effect of deliberately altering the parameters of the chains to yield RC modes that are damped by a simpler, extended environment (490 modes). This leads to qualitative agreement with the *ab initio* model, and could provide an interesting way to benchmark a range of recent theories based on RC master equations **\cite{garg1985effect, hughes2009effective,iles2014environmental}**

2. *“Can the authors make a comment about how easy it would be to make predictions/create designs for optimised singlet fission in other materials using this method?”*

A very interesting point. We see two promising approaches to design and prediction with this method. The first is to attempt to generate fairly indiscriminate high-throughput data on a range of molecules. As discussed above, a fairly modest, single CPU core gives a truly microscopic description of SF dynamics in just a few days; running many structures in parallel should then allow us to rapidly accumulate multidimensional material data. This could be followed by data-correlation analysis of fission outcomes against a wide range of electronic, structural, symmetry or vibrational molecular parameters, as is currently used in many screening approaches to materials discovery. Similarly, data obtained from our approach could – in principle – also be used to train/learn an optimal linear vibronic model for fission in a dimer.

The second, slower approach - pursued in the present MS - is to take molecules with known SF capabilities and try to identify microscopic vibronic processes that promote fission, and which might provide a general strategy for improving fission materials. For DP-Mes, our new energy surface approach revealed the conditions on the vibronic super-exchange dynamics that make fission in DP-Mes efficient (a coherent super-exchange pathway within the *same* environment cluster and a near-adiabatic descent of the excited TT surface through the LE/TT avoided crossing).

The following comment incorporating these ideas has now been added to the conclusions.

” In both cases, access to detailed and accurate dynamics is essential, and we further note that - as the single-core computational costs of our technique allow easy parallelisation - potentially high-throughput simulations could be implemented for dynamics-based screening of such SF materials. We therefore hope that the approach we have set out could provide a useful tool in the emerging drive towards dissipative coherent devices constructed from rationally designed, nanostructured organic materials”

Other comments

1. *“There are a lot of typos and similar small errors in the paper - the authors should do a careful proof read and spell check before re-submission. (A couple of examples - Fig 1 caption mediated not medaited; p4 first paragraph: dimensionality not dimensionally.)”*

We thank the reviewer for pointing out the excessive number of typos in this

manuscript. The document has now been thoroughly proofread and all errors corrected.

2. *“Top of P2, cite further work on using MPS to model open systems e.g. recent paper arXiv:1711.09641.”*

Thank you for pointing out this excellent article (that we had become aware of after the original submission of this article). The published version of this article is now cited prominently in the introductory section and mentioned in the conclusions.

3. *” Fig 1 colour scheme. Red and orange hard to distinguish for the colour blind reader.”*

We have now altered the colour scheme to improve visibility for all readers.

4. *“P2, col. 2: why is the cruciform geometry novel?”*

We thank the reviewer for making this point. The novelty of the geometry is somewhat subjective; indeed, it might be better to say that the geometry is “useful”. Cruciform dimers have been studied in the past, although DP-Mes is the first to be considered in the context of singlet fission. Considering only the carbon skeleton of the molecules, the optimal ground state geometry would be a planar dimer in which all the carbon atoms are conjugated. However, due to the position of the linking c-c bond between the pentacene monomers, repulsive steric effects due to overlap of hydrogens drive the dimer into an orthogonal (cruciform) geometry. In this geometry, the pi orbitals on each monomer have zero overlap by symmetry. Absence of electronic coupling at the cruciform geometry means that we can label the excited states as essentially pure locally excited Frenkel excitons, charge transfer configurations and triplet pairs, allowing easy connection between the diabatic basis we employ in our dynamical simulations and the adiabatic states that are computed in the *ab initio* electronic structure calculations used to build the linear vibronic Hamiltonian. The cruciform geometry also enforces a richer point group symmetry (D_{2d} rather than C_{2v} for a planar structure) that proved very useful for checking the quality of the numerical clustering algorithm that we developed (see below).

The latter points about symmetry are discussed in detail in the electronic structure section of the supplementary information and mentioned in the section on vibronic coupling in the main text (“Interface to TTNS”, p3). We have simply removed the adjective “novel” in relation to the cruciform geometry in the main text to avoid any misunderstandings.

5. *“P3, col 1: why are 252 modes used?”*

We thank the author for spotting this. The full structure of DP-Mes contains 318 atoms, of which 6 normal modes are the trivial translations and rotations. Because of the potential failure of the harmonic approximation for very low frequency modes, and also because their dynamics should not become relevant until after the initial stages of fission, we filtered out modes below $<50/\text{cm}$. Very high frequency modes ($>2000/\text{cm}$) and modes that were very weakly coupled to the dynamics were also excluded. These latter modes were primarily associated with the 40 modes of the non-conjugated side groups that are added to the pentacene molecules to improve the dimer’s solubility in organic solvents. Clarification of this point has been added to the

main text in the section on vibronic coupling and is repeated in the supplementary information (SII). The main text now states underneath Eq. 1:

“To explore these effects, we then employ further DFT calculations to construct a linear vibronic Hamiltonian^{\cite{Kuppel1984}} based on 252 quantum harmonic normal modes \hat{b}_n^\dagger of the dimer's ground state potential surface. This describes the emerging inter-state electronic couplings W_n (bold symbols denote operators in the system Hilbert space) as the system is perturbed from the ground state geometry (see ^{\sm{}} Sec.~II). Note that the number of modes considered in the model is slightly smaller than the total number of modes of the dimer structure shown in Fig.^{\cite{fig:states}} 318. This is due to the removal of low frequency modes ($< 50 \text{ cm}^{-1}$) for which the harmonic approximation is inappropriate, and also the exclusion of the weakly coupled, very high frequency modes associated with the pendant side group atoms. We also found that our results showed only small quantitative changes if we reduced our model to just the 75 modes that account for 90% of the system-bath coupling in each environment cluster (see below), although this reduction does not lead to any advantages in terms of simulation speeds or memory, as shown and discussed in ^{\sm{}} Section IV b. “

As mentioned in the new section **Remarks on simulation performance**, the observation that models containing fewer modes per cluster could produce fairly similar results points to the importance of the ‘reaction coordinates’ in this problem, and may be useful for simpler approaches for this type of problem. This is backed up by our simulations of environments with larger chains to mimic a damped reaction coordinate in the SI.

6. “P3, col 1, paragraph after Eq 1. Can more references to example relevant systems be put after “linear vibronic Hamiltonian is applicable to a very wide range of molecular and supramolecular chromphoric systems,””

“Although for clarity we illustrate our technique with a particular example, our method is designed to deal with linear vibronic models for fairly arbitrary parameter sets. This greatly extends the domain of applications described by Eq. ^{\ref{LVH} \cite{weiss2012quantum}}, and the general form of Hamiltonian in Eq. ^{\ref{LVH}} has been applied to systems as diverse as photosynthetic proteins and the electron-phonon systems of semiconductor heterostructures ^{\cite{van2000photosynthetic,chin2013role,Grosse2008,Lo2012}}. In addition, the model remains unsolved and an important testing ground for new theory. Its rich phenomenology has also been the focus of recent ^{\emph{experimental}} quantum simulations with superconducting and ion trap qubits ^{\cite{potovcnik2018studying,gorman2018engineering}}.”

7. “P3 col 1, next para. I think it should be Fig 2c, not 1c.”

Thank you for spotting this typo, it has now been corrected.

8. “P3, col 2. Can the authors say in the main text how they tell that a minimum of four clusters are needed? And then what determines why 7 are needed for a “faithful” representation?”

The minimum of four clusters is set by the point group symmetry of the molecule (D_{2d}) which has four irreducible representations (irreps). Each normal vibrational mode of

the system belongs uniquely to just one irrep, so in principle the environment can be divided into 4 independent environments. As the matrix elements defining the coupling between electronic states (also labelled by an irrep) mediated by the vibrational displacement must be scalar, i.e. of A_1 symmetry, each group of modes couples to the system through an operator with a specific pattern of non-zero elements that will not, in general, commute with the operators in a different symmetry sector. This is why the minimum number of environments here is four, and this would be achieved if each mode within a cluster coupled to the system (up to a scaling factor) with the same operator. However, it is not guaranteed that each non-zero element of the coupling matrix for each mode in a given cluster will have the same numerical values.

The clustering algorithm handles this by searching for further structure in a given symmetry cluster. In effect, the K-means technique is rather like feature recognition, principal component analysis, or even single value decomposition applied to (operator-valued) data. Here, the algorithm determines the coupling operator that best approximates the couplings of modes in that cluster. This “centroid” operator is rather like an average coupling for the modes that accounts for the largest variance of the data: most of the modes in the cluster couple to the system (up to a scaling factor) through this operator, allowing us to factorise the interaction and implement the chain transformation. However, if there is significant variation in the couplings, not all modes will be accurately approximated by taking this form of the coupling. In this case, these modes can be identified and placed in another environment characterised by another operator, a process that can be repeated until the original distribution is reproduced. The K-means clustering does this by a minimisation of a distance-like measure (in operator space) of the approximate decomposition and the original data, and we used this to determine that seven environments were needed to describe the original linear vibronic Hamiltonian. The need for seven environments becomes especially clear when looking at the 2D projection of the data given in Fig. S3, which has now been further annotated to better explain how the clustering algorithm operates. This is an important point -and a possible research topic in its own right – and we thank the referee for helping us to improve our description of this technique for open systems. We now say in the MS:

“This determines the minimal required set of sub-environments \mathcal{S}_i and their coupling operators $\overline{\mathbf{W}}_i$ that optimally approximate $\overline{\mathbf{W}}_n$ of assigned modes, giving the best decomposition of the system-mode coupling of the form $H_I = \sum_i \overline{\mathbf{W}}_i \sum_n \lambda_n (a_n + a_n^\dagger)$, where the index n runs over the total number of modes in cluster \mathcal{S}_i . The clustering algorithm shows that the DP-Mes couplings require a minimum of four clusters, which is perhaps not so surprising; by group theory, modes should naturally break up into clusters labelled by each 1D irreducible representation (irrep) $A_{1/2}$ and $B_{1/2}$ of the approximate point group D_{2d} of the DP-Mes ground state geometry (see Fig. \ref{fig:states}). This validates our more general clustering approach, suggesting the particular efficiency of clustering for systems with well-defined symmetries. However, there can also be large variances of the coupling patterns around the centroid $\overline{\mathbf{W}}_i$ within a given symmetry class. For Dp-Mes, the clustering algorithm determines that seven clusters are needed for a faithful representation of the original \textit{ab initio} model, which can be seen clearly in the visualisation of the coupling clusters presented in \sm{} Fig.~3. To the best of our knowledge, K-means clustering has never before been applied to the open system problems, and exploring the formal aspects of this type of analysis may generate interesting insights of comparable utility to the theoretical properties of reaction coordinate and chain transformation modes. “

1. "P3, col 2. The authors associate the clusters with the group irreps - but in Fig 1 A₁ doesn't appear. Why is this?"

We thank the reviewer for spotting this. The A₁ modes cause an energy (Stokes) shift for all electronic states and do not couple different electronic states (diagonal coupling). They do play an important role in dynamically tuning the relative energies of the states during dynamics, and so must be included in the simulations. However, in order to not over complicate Fig. 1b, we decided to show only the interstate transitions mediated by the modes, as needed for our later discussion (hence the absence of A₁). We have added this sentence to the caption of Figure 1 to clarify this point.

"A₁ modes shift the energy levels of each state but do not induce any coupling, and have been omitted from the figure for simplicity. However, it shall be shown that they do play an important role."

2. "P3, before Eq 2. 1c to 2c again?"

Thank you for spotting this, it is now corrected.

3. "P4, col 1. How do the authors know that $n_b > 100$? There is also a missing reference in this paragraph."

This is based on our convergence tests w.r.t n_b . For convergence to the level of ~3% we found that $n_b \sim 100$ was required. Some justification of this is already suggested in the results, which show that several reaction coordinate modes contain on average 3-4 quanta (but need many more levels to give an accurate description of their entangled wave functions). More details on convergence are given in Supplementary Section IV. Thank you for spotting the missing reference. This has now been added and refers to a review on entanglement area laws in quantum-correlated systems by Eisert *et al.* (Rev. Mod. Phys. 82, 277, 2010).

4. "P4. In discussing the linear scaling of MPS, I think it is important to say that this relies on a constant/ reasonably bounded D , obtained using SVDs and subsequent truncation."

A very good point, we have changed our discussion to point this out to the reader. We now say:

"Thus, for a fixed bond dimension that is sufficiently large to capture the entanglement appearing over the dynamics, this type of ansatz effectively creates a linear-scaling problem in system size. However, this becomes a higher-order polynomial (cubic) when including the overheads for tensor contractions, the sequential updating of each tensor and the extraction of observable quantities when simulating the dynamics."

5. " Fig 6. Why are labels d and e ?"

Another unfortunate typo (this figure was previously packaged in a larger multi-panel

figure). We offer our apologies for the embarrassingly high error-rate in this MS. The figure has been corrected.

Reviewer Two

While we strongly disagree with the referee's view that this work lacks the "clear advance" needed for publication in Nature Communications (see below), we would like to begin our response by thanking the referee for the time that he or she has taken to review and report on our manuscript. In particular, we are very grateful for his/her observation that we accidentally referred to MCTDHF. We are aware of the relevance and literature of ML-MCTDH, and the references that we included in our MS were in fact about this technique. Unfortunately, a typo in the MS mistakenly referred to these references as (fermionic) MCTDHF, which we agree is not comparable with the types of model explored in this paper. Indeed, since starting this project we have had many fruitful interactions with researchers working with ML-MCTDH, and we hope to explore the links between these techniques in the future.

The referee also gave several important references related to low-rank tensor approximations, variational principles and tensor dynamics from the applied mathematics literature. Some of these were in fact cited in our original manuscript's supplementary information, and have indeed been inspirational to us in the development of our approach to open systems. We have now expanded the introductory passage of our MS to ensure that these historically important contributions are given their proper place alongside the physics and chemistry literature in the main text.

The referee correctly points out that time-dependent variational principles have previously been applied to compute MPS/tensor train dynamics (not least by some of the present authors) and that linear vibronic dynamics have previously been investigated with techniques such ML-MCTDH. However, we feel that the referee has perhaps misunderstood the aims and scope of this article, and has consequently overlooked many of the new developments that - we strongly believe - constitute a very clear advance in the simulation of open quantum systems. These simulations ultimately underpin the understanding of complex dissipative dynamics in molecules and other systems (see the new list of possible applications added at the suggestion of Ref 1).

The simulation and understanding of open quantum systems is a fundamental area of research in physics and chemistry, with profound implications for the exploitation of quantum phenomena in future technologies. As we set out in our MS, the microscopic wave function approach that we pursue must include a sufficiently large number of environmental modes to generate irreversible/dissipative dynamics (over suitably long times). This has been done in ML-MCTDH in several toy models where the environments are coupled in a simple way to the electronic system, with the interesting physics relating more to the perturbed dynamics of the electronic/system degrees of freedom. Here, we explore the relatively ill-understood case where complexity in the environmental dynamics drives the electronic phenomena (here, fission); indeed, there are no electronic dynamics in our orthogonal dimer in the absence of the environments. The complexity and richness of the dynamics arises from symmetries and

structures in the fully *ab initio* model parameterised by DFT methods, as well as the fact that we deal with a very large number of entangled environmental quantum states.

To be able to treat this type of model – which is essential for understanding a broad range of photophysical systems – it is not possible to use a simple implementation of established MPS/tensor train methods. The referee refers to our work as “numerically sophisticated” - which it is – but this seem to gloss over the actual physical insights and new advances we have had to develop in order to treat this problem. In particular, both Referees 1 and 3 made mention of the key use of machine learning clustering and the introduction of entanglement renormalisation analysis that enable this simulation to be run with fairly modest computational resources. Understanding the “entanglement” topology of open systems could be a powerful way to understand or manipulate decoherence, and this is particularly highlighted in the present work by the 6 orders of magnitude compression in memory requirements that this analysis provides (with the addition of ER tensors). We note that our method of constructing the tensor network is likely to have profound connections with the building of “balanced trees” in ML-MCTDH, and we hope that the entropy measure we use to construct our tensor network might play a useful role there.

More generally, there is considerable interest in the physics of systems coupled to multiple environments, particularly in the fields of nanoscale heat engines, non-equilibrium many body states, and hybrid quantum devices (esp. quantum sensing). In many cases, only one environment is given a fully quantum mechanical treatment and the others (normally one) described by master equations (often Lindblad theory). The present work treats seven environments, and shows how the interplay of their concerted dynamics drives the fission process. As mentioned in our list of wider applications, our technique is fit to explore this nascent field of multi-environment open systems.

The use of K-means clustering methods to gather the vibrational modes into a few independent environments has never before been proposed or implemented in the context of open quantum systems (or any other quantum system, to the best of our knowledge). The convergence data in the supplementary materials shows that transforming the environments into simple 1D chains leads to rapid convergence w.r.t. the chain bond dimensions, making the cluster step of particular importance. Again, one of our goals was to be able to bring TTNS/MPS/TDVMPs methods to the problem of realistic molecules rather than toy models, and the clustering scheme we introduce becomes increasingly important as the irregularity of the coupling matrices in the vibronic Hamiltonian becomes large, as they are likely to when extracted from DFT applied to real structures.

We have also exploited the structure of the tensor system-environment wave function to introduce a new way to visualise the dynamics of singlet fission on effective energy surfaces that track the most important environmental configurations (coordinates). This, again, has never been presented for an open system and, here, allows us to pinpoint not only the mechanism, but also the individual events leading to efficient fission. Crucially, these events involve non-occupied electronic states that would be invisible in any master equation approach, or any wave function approach that only focused on electronic or vibrational observables.

We believe strongly that the innovation rate in our MS is very high and will encourage new work with TTNS dynamics in a number of fields across physics and chemistry (we are already aware of a few). The feedback we have had from peers - and, indeed, two referees - has also been very positive, particularly regarding the the fundamental idea that we have shown to be possible an approach that was previously considered impossible. We therefore feel that we have met

the “clear advance” requirements for publication in Nature Communications in several directions, and hope that this response has now made this clearer.

Reviewer Three

We thank the reviewer for closely reading our manuscript and for the very insightful and encouraging report. We are especially grateful to the reviewer for commenting on the novelty of our numerical approach, and also for their recommendation to publish our article subject to minor revisions. In response to the referee’s comments:

1. *“I do not agree with the comment ... MPS are only efficient for 1D systems.....Of course due to linear topology 1 D systems with nearest neighbour interactions are optimal for MPS. I suggest to modify the text MPS are mostly efficient for 1D systems..... “*

An excellent point. We have now made this change in the main text.

2. *“The bibliography w.r.t to tensor networks is almost appropriate. I would like to add the monograph of W. Hackbusch.”*

This very useful reference has been added to the introductory section of our main text.

3. *“I have not looked for typos; but i have seen at least one:”*

Thank you for spotting this typo. A number of other typographical errors have now been corrected.

4. *“The paper is dressed to experts in physics. The various definitions are hard to understand from an abstract point of view. To get a wider audience it seems to helpful to put more effort on the mathematically clear and precise definitions”*

We have tried to find a compromise between the formal and informal presentation of the ideas in the main text, and appreciate that the formal aspects of this work could be of interest for the mathematical physics/applied physics community. However, we feel that the appropriate place for the lengthy and formal mathematical developments underlying our method is in the supplementary information and references, allowing more space in main text for the discussion of singlet fission and more intuitive ideas that could trigger new applications across physics and chemistry. We hope that the referee will agree that the Supplementary Information clarifies the rigorous mathematical and computational (algorithmic) underpinnings of our results, and that any readers seeking a more abstract understanding of our approach will be able to find one in our SI and references.

REVIEWERS' COMMENTS:

Reviewer #1 (Remarks to the Author):

I would like to thank the authors for their detailed response to my report, which I have read carefully and am satisfied with. I am therefore now happy for the paper to be published.

I would however request one further clarification in the text. When comparing to the TEMPO method in the response, the authors make it clear that the comparison is made at a non-critical point ($\alpha=0.5$). Could this be stated also in the main text?

Brendon Lovett, St Andrews University

Reviewer #2 (Remarks to the Author):

First, I have to admit that I am not familiar with the criteria Nature Communications used to accept paper. Previously I was merely reading the editor's comments and thought they made sense. I admit that I might have been set the bar too high. As I said, I think the method is sophisticated. I am also aware of a couple of chemists' work on singlet fission using ML-MCTDH. This paper is certainly better.

Although I still don't agree with most of the authors' response regarding the comparison of the current method with ML-MCTDH (to reviewer 1 and me), I would not use this to reject the paper. In the future I would be happy to discuss with the authors on the technical points, and it is entirely possible that I am wrong.

I would raise a few minor points for the authors to consider.

1. The authors use ref. 13 to point out the difficulty of applying ML-MCTDH. However, the authors of ref.13 were inexperienced users of ML-MCTDH (offered in the Heidelberg package as a blackbox, and there are other ML-MCTDH codes too). I would not infer too much from their results. It is typical to use binary trees (hierarchical Tucker form) or skewed trees (tensor train), but it is merely a matter of convenience. If the authors have a systematic way, that would be great.

2. When comparing the efficiency with ML-MCTDH, the authors cite ref.24 and state "we believe that our algorithm should be broadly comparable to ML-MCTDH in terms of resources and speed." Ref.24 was a somewhat older work (year 2008). There are some other studies done using ML-MCTDH, for example, see the recent review: H. Wang, J.Phys.Chem.A, 119, 7951 (2015) and references therein. In particular, check Fig.1 and 2 of that paper. Fig.2 has a result of the spin-boson model with 10,000 modes. I would think that the Hilbert space is larger than $10^{\{500\}}$ the authors put. On the other hand, the Hamiltonian in the current manuscript is more complex, so I would guess that the current work is more challenging.